# Recent Developments in Rodent Models of High-Fructose Diet-Induced Metabolic Syndrome: A Systematic Review

**DOI:** 10.3390/nu13082497

**Published:** 2021-07-22

**Authors:** Alvin Man Lung Chan, Angela Min Hwei Ng, Mohd Heikal Mohd Yunus, Ruszymah Bt Hj Idrus, Jia Xian Law, Muhammad Dain Yazid, Kok-Yong Chin, Sharen Aini Shamsuddin, Yogeswaran Lokanathan

**Affiliations:** 1Centre for Tissue Engineering and Regenerative Medicine, Faculty of Medicine, Universiti Kebangsaan Malaysia Medical Centre, Jalan Yaacob Latif, Kuala Lumpur 56000, Malaysia; alvinchanmanlung@outlook.com (A.M.L.C.); angela@ppukm.ukm.edu.my (A.M.H.N.); ruszymah@ppukm.ukm.edu.my (R.B.H.I.); lawjx@ppukm.ukm.edu.my (J.X.L.); dain@ukm.edu.my (M.D.Y.); s.sharenaini@ppukm.ukm.edu.my (S.A.S.); 2Ming Medical Sdn. Bhd., D3-3 (2nd Floor), Block D3 Dana 1 Commercial Centre, Jalan PJU 1A/22, Petaling Jaya 47101, Malaysia; 3Department of Physiology, Faculty of Medicine, Universiti Kebangsaan Malaysia Medical Centre, Jalan Yaacob Latif, Cheras, Kuala Lumpur 56000, Malaysia; mohdheikalyunus@yahoo.com; 4Department of Pharmacology, Faculty of Medicine, Universiti Kebangsaan Malaysia Medical Centre, Jalan Yaacob Latif, Cheras, Kuala Lumpur 56000, Malaysia; chinkokyong@ppukm.ukm.edu.my

**Keywords:** high fructose, metabolic syndrome, syndrome X, metabolic syndrome X, rodent, rat, mice, mouse

## Abstract

Metabolic syndrome (MetS) is the physiological clustering of hypertension, hyperglycemia, hyperinsulinemia, dyslipidemia, and insulin resistance. The MetS-related chronic illnesses encompass obesity, the cardiovascular system, renal operation, hepatic function, oncology, and mortality. To perform pre-clinical research, it is imperative that these symptoms be successfully induced and optimized in lower taxonomy. Therefore, novel and future applications for a disease model, if proven valid, can be extrapolated to humans. MetS model establishment is evaluated based on the significance of selected test parameters, paradigm shifts from new discoveries, and the accessibility of the latest technology or advanced methodologies. Ultimately, the outcome of animal studies should be advantageous for human clinical trials and solidify their position in advanced medicine for clinicians to treat and adapt to serious or specific medical situations. Rodents (*Rattus* *norvegicus* and *Mus* *musculus*) have been ideal models for mammalian studies since the 18th century and have been mapped extensively. This review compiles and compares studies published in the past five years between the multitude of rodent comparative models. The response factors, niche parameters, and replicability of diet protocols are also compiled and analyzed to offer insight into MetS-related disease-specific modelling.

## 1. Introduction

### 1.1. Metabolic Syndrome

The prevalence of obesity, diabetes, and cardiovascular diseases in modern society has been a global problem since the past century, and is still growing. Although “metabolic syndrome” (MetS) was coined in the early 1940s, the topic only became known after the works of Vague et al. and Haller and Hanfield, who successfully correlated the prevalence of obesity to diabetes and the hallmarks of MetS [1,2]. On 1 April 2020, the World Health Organization (WHO) reported that the prevalence of obesity had tripled over the four decades between 1975 and 2016 [3]. It was heavily implied that the low-to-mid-income countries are greatly affected by this and have since shown more linked deaths than underweight issues. Up to 1.9 billion adults (25% of the global population) have overweight problems. Higher and no variations between males (39%) and females (40%) have been reported. In the same year, the International Diabetes Federation (IDF) also reported that 223 million adults aged 20–79 years have diabetes [4]. This projection is expected to peak at 700 million by 2044. Both organizations have also confirmed that the prevalence of obesity and diabetes has increased in children or adolescents, which is a major concern.

MetS is the phenomenon of accumulated symptoms that complementarily and progressively deteriorate the person’s wellbeing. The contributing factors may vary between each person’s exposure (environmental) or susceptibility because of hereditary (genetics) traits. Among the MetS contributors are unhealthy habits, sedentary lifestyle, poor diet choices, family history, socioeconomic status, and education [5]. Habits such as excessive alcohol consumption can cause hepatic dysfunction from the constant liver output of detoxifying metabolites. Smoking and narcotics are also common habits that deteriorate pulmonary and cardiac functions. A sedentary lifestyle is the privilege of access to advanced technology and entertainment that involves less physical stress and engagement. This drives the manifestation of obesity and the lack of mitochondrial stimulus for efficient energy production. By far, poor diet choices are the greatest contributor to MetS of the 21st century [3]. Previous correlations of the socioeconomic relationship of a person’s income to obesity have shifted greatly, as recent findings show larger statistics in low-to-mid-income populations. The ease of accessibility to cheaper and hypercaloric diets driven by global franchises, as well as the incorporation of high salt, fat, and carbohydrate into traditional or commonly accessed foods, has seen to the growth of obesity. The traits of these nutritional imbalances can be defined as the Western diet phenomenon [6]. Conversely, the higher-income population has opted for a healthier and more organic lifestyle, which is not viable for the low-to-mid-income populations. Putting aside hereditary diseases such as type 1 diabetes, the role of genetics is less impactful, as it can be simplified as the person’s susceptibility or tolerance of biochemical alterations. It is difficult to quantify the tolerance level, as it varies even among siblings, but can be controlled when a balanced diet and lifestyle are provided [7]. Lastly, education is better interpreted as self-awareness of healthy choices, habits, and the other factors as listed above. It is ultimately the person’s discipline and restrained use of enriched resources to mitigate the effects of MetS.

The characteristics of MetS are high body mass index (BMI), hyperglycemia, hypertension, dyslipidemia, and insulin resistance. A joint interim statement from the IDF Task Force on Epidemiology and Prevention; National Heart, Lung, and Blood Institute; American Heart Association; World Heart Federation; International Atherosclerosis Society; and the International Association for the Study of Obesity states that a diagnosis of MetS is accepted if at least three of the five characteristics above are present [8]. A set of these conditions can lead to the development of major metabolic diseases such as cardiovascular diseases, type 2 diabetes, non-alcoholic fatty liver disease (NAFLD), osteoporosis, cancer, and death. Hence, the importance of studying MetS is to serve as a preliminary measure before the development of chronic disease. The present review was aimed at performing a qualitative analysis on the recent development of MetS in rodent models. It is intended to highlight the criteria for the successful establishment of an animal model of MetS. It is also aimed at compiling and comparing the test parameters for MetS and related diseases. Rodent models have been used extensively since the 18th century until today as the most preferred model for animal studies. Rodents are considerably easier to manage compared to larger mammalian families such as leporids, swine, equines, or primates. The selection of rodents is also driven by the ease of sourcing the animals; their lifespan, which is suited to the average study duration; and the complete mapping of their genetics and pathophysiological characteristics.

### 1.2. Fructose as a Dietary Choice

The present review focuses on diet-induced obesity (DIO) for simulating the development of obesity as a result of poor diet and lifestyle choices. One hallmark of DIO is the exaggerated incorporation of sugar, fat, and salt in the diet. References thereof are known as Western diets, introduced from regions that actively encourage chemical additives in processed foods, fast food franchises, and carbonated sugary beverages [9]. For example, the incorporation of fructose in beverages and processed food has been noticeably increasing. This popularization stems from fructose yielding a sweeter flavor, and it is most often added supplementary to generic sugar, i.e., sucrose, which is itself a disaccharide compound of glucose and fructose [10,11,12]. Fructose is one of three common sugars (monosaccharides) but is not directly processed in most metabolic processes such as energy (ATP) generation. Instead, fructose aids liver synthesis of glycogen molecules through a series of steps that overlaps with gluconeogenesis. In the context of MetS, fructose enhances the synthesis of triglycerides (TGL) from glycerol and fatty acid (FA) formation. Subsequently, TGL is stored as fat until a depletion of deposited glucose triggers a negative feedback. However, as glucose scarcity is less probable, the metabolized fructose is deemed in excess and is stored principally as fat. Evidence has been presented indicating that fructose bears similarity to narcotics, enabling unhealthy compulsion and downstream hyperphagia [9,13,14]. In this context, high fructose consumption is associated with MetS prevalence.

### 1.3. Fructolysis

The machination of fructolysis begins through the diffusion of fructose by the transporter GLUT5 in a concentration-dependent manner. Aided by GLUT2, it crosses the intestinal lumen walls to exit to the bloodstream. As blood courses throughout the body, the fructose is bound by the GLUT2 transporters of the liver [11,15,16]. Therein, fructokinase catalyzes fructose into fructose-1-phosphate (F1P). In turn, F1P is cleaved by aldolase B into dihydroxyacetone phosphate (DHAP) and glyceraldehyde (Figure 1). Subsequently, DHAP is converted into glyceraldehyde-3-phosphate (G3P), pyruvate, acetyl-CoA, and FA molecules. Meanwhile, glyceraldehyde is first converted into glycerol, then glycerol-3-phosphate (Gro3P). Finally, TGL is formed following Gro3P and FA esterification. TGL and FA are both released into the bloodstream, contributing to dyslipidemia, which is defined as abnormal lipid levels in the blood [17,18].

Between the course of pyruvate conversion to acetyl-CoA, citrate, CO_2_, and ATP molecules are released. Citrate and ATP act on the phosphofructokinase (PFK) enzyme (of the glycolysis mechanism), responsible for phosphorylating fructose-6-phosphate (F6P) into fructose-1,6-bisphosphate (FBP). ATP activates PFK to stimulate functions while citrate inhibits it via the PFK affinity for ATP binding. The pathway of fructolysis is able to circumvent this crucial regulatory process, which is a devastating concern since there are absent or minimal downstream homeostatic mechanisms that are able to reverse or attenuate said impact [9,10,11,14,15]. Thus, its effects widely impact many physiological systems, with potentiating acute to chronic levels of damage.

As mentioned above, the primary product of fructolysis is TGL. Among the eventual outcomes of metabolic disorders, the damage would likely stem from the organs responsible for blood circulation and filtration. Hence, the liver, kidney, pancreas, and vascular system are the earliest victims of chronic fructose ingestion [15,19]. Excess adipogenesis and lipogenesis will precipitate increased adipocytokines, such as leptin. Although leptin balances energy homeostasis through hunger, overproduction will accumulate as leptin resistance. Consequently, this leads to hyperphagia, obesity, and increased percentage of fat mass [19]. Adipocytokines also consist of inflammatory cytokines. The commonly referred MetS cytokines are tumor necrosis factor alpha (TNF-α), interleukin-6 (IL-6), and reactive oxygen species (ROS) or antioxidants. The fluctuating production of these metabolites can lead to inflammation, insulin resistance, and hypertension [20,21,22,23]. Individually, each of these adipo-secretomes exerts mild effects, but may collectively result in one or more systemic complications.

### 1.4. The Bone and MetS

In Figure 1, the focus of MetS is centered on insulin resistance and/or inflammatory cytokines. Therefore, the categories of liver, cardiovascular, renal, and oncological diseases are the highlights. The inclusion of bone diseases is an extremely intriguing premise because of the condition that ageing and sex-favoring disease are minimized. Osteoporosis has significantly greater incidence in women than in men, more so with ageing co-factored simultaneously. Contradictory to current beliefs, men and young adults of both sexes have osteoporosis risk derived from severe MetS [24,25]. Although MetS has been reported with non-significant values, it has been noted that increased weight gain, adiposity, and oxidative stress are nevertheless potential contributors [26,27,28,29]. As the topic is highly appraised, the notion of bone-related diseases experiencing a paradigm shift by MetS is a major concern for other factor-limited diseases. That the expanding efforts in MetS will provide further clarity is of interest.

### 1.5. The Brain and MetS

There have been equal postulations of MetS in relation to cognitive decline and unhealthy pressure on neurological diseases. Globally, two of the most common age-related metabolic deteriorations of the brain are Alzheimer disease and Parkinson disease. Several authors have explored the impact of disproportionate inflammatory secretions and oxidative reduction on accelerating or aggravating the encephalic system [30,31]. Insulin resistance and irresponsive receptors in the brain significantly decrease blood flow, leading to chronic oxidative stress and damaged cognitive roles [30,31]. However, this premise is challenged due to insufficient evidence and significant values [32]. Based on these studies, minimal to absent efforts in genomic and proteomic studies are apparent, although this may be an issue of accessibility to advanced methodology or technology during the past decades. Hence, this is another research topic to be rediscovered. If there is much evidence, novel studies could pave the foundation in neurology and gerontology towards novel therapeutics for neurodegenerative and mental disorders.

## 2. Materials and Methods

The methodology for this systematic review was approved and registered under the guidelines of PROSPERO (International prospective register of systematic review, PROSPERO ID: CRD42021238988).

### 2.1. Keyword Selection

To identify the keyword validity and synonyms, “Metabolic Syndrome” and “High Fructose” were searched through medical subject headings (MeSH), a PubMed vocabulary thesaurus. The keyword search results served as standard search terminologies for articles.

### 2.2. Database Selection and Result Filtering

The Scopus, PubMed, and Web Of Science (WOS) databases were selected from available access provided by the National University of Malaysia Faculty of Medicine. The keywords were searched and the databases were filtered specifically for “research articles” or “journal articles” published in 2016–2020 (5 years). The bibliographies were downloaded from the respective databases and labelled appropriately (e.g., PUBMED_12_11_2020 109 results).

### 2.3. Inclusion of Titles, Abstracts, and Keywords

The bibliographies were uploaded to citation software (Mendeley). Files were downloaded separately, compiled in a different folder, and duplicates were combined. Duplicates were removed automatically but were also removed manually for assurance. The titles and abstracts were initially screened according to the inclusion and exclusion criteria. The inclusion criteria were: “high fructose” and “metabolic syndrome” and ≥3 MetS symptoms: (i) Increased weight or abdominal circumference, (ii) dyslipidemia, (iii) hypertension, (iv) decreased high-density lipoprotein (HDL) levels, and (v) hyperglycemia. The exclusion criteria were: (i) long-term study or aging effect, (ii) generational study, (iii) absent metabolic disorders or parameters, (iv) non-fructose diet, and (v) high fructose in combination with other diet.

### 2.4. Inclusion of Articles Based on Methods and Results

A second screening was performed by evaluating the method for fructose-only MetS induction. For example, the combination of high-fat, high-salt, and high-carbohydrate diets with fructose was excluded. Next, the results of individual studies must also have presented a valid comparison of fructose diets to control diets before any treatment. All excluded and included studies were retained and numbered for totals (Figure 2).

Figure 2 shows that the initial search yielded 109 articles from PubMed, 361 articles from Scopus, and 256 articles from WOS. In total, the search yielded 726 articles when combined. Duplicate auto-removal was performed upon combination, followed by manual duplicate removal. The total pooled database numbered 597 individual articles, and 129 duplicates were removed. Two reviewers individually filtered the articles following the agreed parameters. Although both reviewers had 18 articles each, four articles were not identical. Post-discussion, a total of 18 articles were agreed for further review. The articles reviewed are presented as below (Table 1), in chronological years (earliest to recent).

## 3. Results and Discussion

### 3.1. Rodent Selection Criteria

#### 3.1.1. Susceptibility of Specific Rodent Strains

Rodent selection is an important factor in simulating MetS in an animal model. Species and sex variability should be controlled based on the requirements of the experiment. In such studies, there is notable inter- and intra-species variation in the outcome between mice (*Mus musculus*) and rats (*Rattus norvegicus*). Among rats, the most reliable strains are Sprague–Dawley (SD) rats and spontaneously hypertensive rats (SHR) [33,34,35,36,38,39,40,41,43,46]. Both strains can manifest the many characteristics of MetS with no serious inconsistencies or conflicts. SD rats and SHR have been established by decades of research and are considered useful models of DIO [47,48,49]. However, the Wistar rat model has a recurring issue. Table 1 shows that the Wistar rat models demonstrate multiple reported complications in developing weight gain and hyperglycemia [21,33,38,44,45]. Wistar rats remain potential models, although not exclusively predisposed for DIO; perhaps they can be categorized towards a “generalized model”. Only one study reported hyperglycemia and hypertriglyceridemia with abdominal fat mass of greater impact in Wistar rats as compared to SHR [36]. However, this may have been due to the shorter feeding period.

For mouse models, C57BL/6J mice are the *M*. *musculus* subjects of the highest frequency [20,42]. As per the test parameters, physiological and biological indicators do not pose issues for establishing MetS. One of the seven selected articles reported that C57BL/6 mice did not display weight gain and hyperglycemic levels, a reflection of Wistar rats. Astonishingly, C57BL/6J mice did not lead in a study on the genetic comparison of MetS in three different species. In fact, DBA/2 J (DBA) mice displayed a greater affinity for developing MetS over C57BL/6J mice [42]. However, strain specificity is not the sole factor, as the absence of weight gain and hyperglycemia levels may again be factored by the dosage and duration of the high-fructose diet, as discussed later.

#### 3.1.2. Sex as a Limitation of MetS

Table 1 shows that only a single study had included female subjects. The authors deduced that the presence of biological female organs and hormones plays an important role in protecting the body from MetS symptoms. Compared to non-ovariectomized female rats, OVX rats were at a relative disadvantage from the high-fructose diet, although still lacking in developing MetS compared to the male rats [34]. The outcome of that study only further implicates the use of male rats as sex-controlled models. The protective role of the ovarian hormone, estrogen, preserves the metabolic status of reproducing female individuals. The role of estrogen includes shedding excess adiposity, regulating insulin-mediated glucose and lipid metabolism, and reducing hyperphagia from behavioral changes and stress levels [50,51]. To strengthen this point, various studies on menopausal symptoms, where estrogen production is decreased, have noted the propensity of female subjects to accumulate abdominal fat mass, dyslipidemia, hyperglycemia, hypertension, and inflammation, the prime candidates for obesity and other health risks [51,52]. Hormone-driven satiety changes and increased dietary intake have also been reported from menopausal stages that mediate the development of these symptoms.

### 3.2. Test Parameter Optimization and Selection

#### 3.2.1. Physiological and Biochemical Parameters

Referring to Table 1, the common physiological and biochemical parameters reflect the categories of symptoms established under the definition of MetS; they are: increased body weight (BW), hypertension, hyperglycemia, dyslipidemia, and insulin resistance [8]. The commonality of including caloric intake (hyperphagia) is subjective, as non-significant outcomes have been reported. The table shows that there were no correlations or patterns, but random chances when factored by fructose concentration and duration [20,21,34,38,44,45,46]. The same randomness has been observed in the outcomes for organ or tissue mass [21,22,23,33,34,36,37,38,39,42,44,45]. However, the collection of complete organs or specific tissue mass is highly encouraged for further analysis, as described below.

#### 3.2.2. Additional Parameters of MetS-Related Diseases

Table 2 lists the objectives of the individual studies and their respective test parameters (as additional parameters). The purpose is to share the potential of several selected parameters that may be considered in the future for establishing and studying MetS models. Most MetS studies were not performed independently but were supplementary to other diseases. Two outstanding candidates are the inclusion of histological staining and inflammatory markers. The inflammatory markers that are significantly impacted and are useful indicators are: C-reactive protein [37], TNF-α, IL-6 [20,21,22,23], IL-1β [22], and IL-10 [20,22]. Table 1, Table 3 and Table 4 show that several hallmarks of MetS were not achieved, namely BW, hyperglycemia, and hypertriglyceridemia [21,33,38,44,45]. Through histological findings, these studies were able to identify significant changes at an anatomical level, even though visible but non-significant alterations were reported by the biochemical markers [23,36,37,44]. Signs such as liver fibrosis, abdominal fat deposition, and vascularization were clear. Despite its inclusion, histopathological analysis could only confirm the presence of changes, but was unable to quantify their severity. Incorporating higher-sensitivity analysis, for example, genetic and protein expression analysis, would refine the significance of the results. These methods would enable detailed quantitative and qualitative analysis of the imbalanced metabolites contributing to MetS [20,35,39,42,46]. In brief, complex analysis through advanced machination can provide precise diagnoses that may advance medical care towards personalized medication and novel therapeutics.

### 3.3. Duration and Dosage of High-Fructose Diet

#### 3.3.1. Period of MetS Induction in Animal Models

The 6-week duration was the protocol duration with the greatest frequency in studies on MetS induction (Table 3). Additionally, the outcomes of the six 6-week studies achieved all categories of the MetS physiological and biochemical parameters listed in Table 1 [22,34,36,41,43,45]. Logically, prolonged induction of a MetS diet would have common outcomes, as it is described as a chronic induced syndrome, deteriorating over time. However, Table 1 and Table 3 show that prolonged duration does not achieve several factors such as body mass, organ or tissue mass, and hyperglycemia. Another confounding variable is that species, sex, and fructose diet dosage were not considered. Upon cross-factoring, the outcomes of these three factors were identical to that of studies discussing species differences, specifically Wistar rats [21,33,38,44,45]. This highlights the importance of strain selection and the predisposition of the strains to diet-induced obesity and MetS symptoms.

#### 3.3.2. Concentration of Fructose Diet for Inducing MetS

A high-fructose diet is a part of the triad that includes species and duration for inducing MetS. The selection of fructose concentration is often an overlooked preparation, as intake in rats relies more on palatability and ease of access over satiation levels [53,54]. These special diets are noticeably difficult to solidify and may be unappealing to the rats if produced in private or non-commercially. However, fructose dissolved in water did not pose difficulties, as hydration has greater importance over hunger. The natural ability of rats to resist hunger and persist in harsh environments differs from that of humans and may be a confounding variable by limiting the induction of MetS via DIO [9,54]. Table 4 shows that the most reliable concentrations for solution- and pellet-based high-fructose diets were 40% and 66%, respectively [41,43]. These two studies were able to achieve all MetS determinant parameters, although they did not incorporate all aspects thereof, such as organ or tissue mass, caloric intake, blood glucose levels, and insulin resistance. However, Table 1 shows that a 60% fructose-based diet encompassed all categories of MetS and achieved all but one: increased BW, length, or abdominal circumference [21,35,39,40,45]. Of those five articles, only one reported visibly increased BW, but it was not statistically significant [35]. However, this challenge from the individual study could be subjective; hence, 60% fructose pellets might be deemed equally competent to the single-study 66% fructose pellets [41]. Arguably, another factor to consider is the price of pellets, which is much higher than that of crystalline fructose. These special diet pellets can be made with economically viable ingredients, although this may require further optimization, while purchasing commercial diets may not be a sustainable cost for long-term studies. Crystalline fructose in water is, however, an inexpensive and manageable source.

This review was performed emphasizing a fructose diet exerting physiological and biochemical changes that lead to MetS manifestation. An abundance of experimental protocols utilize high-carbohydrate diet, high-fat diet, and sucrose water (or a combination thereof). Whether these are superior to a pure-fructose diet has not been reviewed here and may serve as an alternative. Carvajal et al. reviewed different diets for inducing MetS in 2020 [9]. Their review could serve as a reference for analyzing the nutritional properties of single or combination high-fat, -sugar, and -salt pellets. Other aspects have not been discussed, such as MetS models in other species. Although rodents are extensively used in scientific research, it is likely due to less restrictive ethics approval. Other mammalian models, such as felines, canines, leporids, swine, equines, and hominoids, may not benefit from this review. Hence, we suggest selecting from the available animal models before proceeding to protocol induction and optimization.

## 4. Conclusions

The best represented rodent species are SD rats and/or SHR and C57BL/6J mice. Ideally, the rodent subjects should be male to exclude the protective nature of female hormones against MetS development, or sex may otherwise be justified in female-specific studies. A study duration of ≥6 weeks is optimal. The fructose concentration is dictated by the method of feeding, which varies with the researcher’s ability to produce or procure pellets. Otherwise, 40% fructose water or 60% fructose pellets yield the best outcomes. Lastly, MetS is often studied in parallel or related to selected diseases. General parameters should encompass MetS prognosis of obesity, hypertension, hyperglycemia, hypertriglyceridemia, dyslipidemia, and insulin resistance. Here, we have tabulated the specific study parameters factored by MetS, and these parameters should be considered when planning studies involving MetS.

## Figures and Tables

**Figure 1 nutrients-13-02497-f001:**
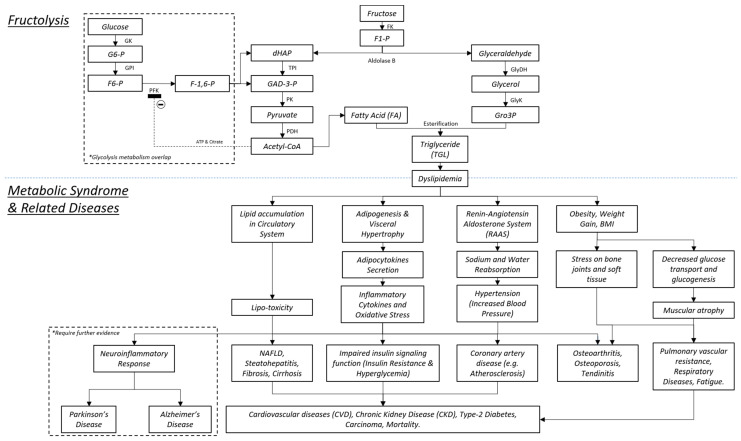
Fructose enzymatic process and physiological impacts to achieve MetS and related diseases. Abbreviations: GK, glucosekinase; G6-P, glucose-6-phosphate; GPI, phosphoglucose isomerase; PFK, phosphofructokinase-1; F-1,6-P, fructose-1,6-phosphate; dHAP, dihydroxyacetone phosphate; TPi, triophosphate isomerase; GAD-3-P, glyceraldehyde-3-phosphate; PK, phosphofructokinase; PDH, pyruvate dehydrogenase; FK, fructokinase; F1-P, fructose-1-phosphate; GlyDH, glycerol dehydrogenase, GlyK, glycerate kinase.

**Figure 2 nutrients-13-02497-f002:**
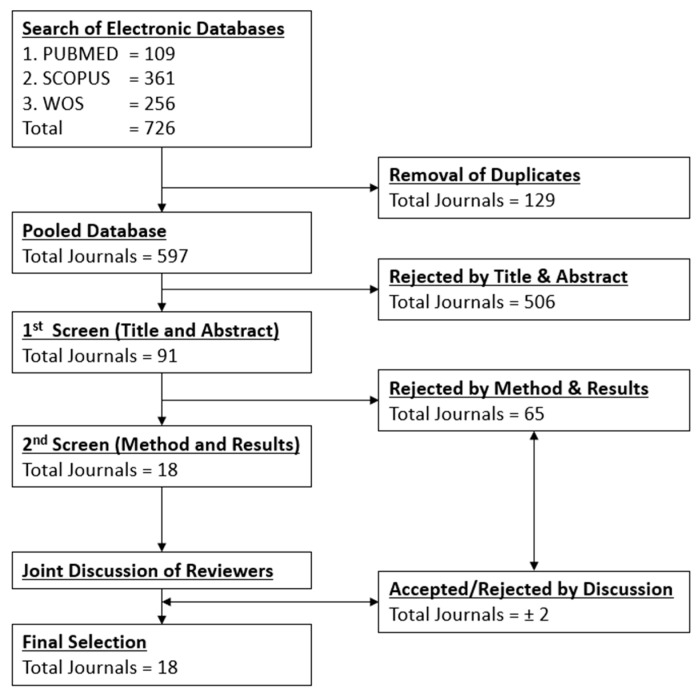
PRISMA flow diagram for process of article selection from pooled database of PUBMED, SCOPUS and WoS.

**Table 1 nutrients-13-02497-t001:** Main factors attributed to MetS establishment in rodent models.

No.	Author and Year of Publication	Rodent Species/Subtype, Sex, Age, Weight Sample Size (n) * n/a = Undisclosed	Fructose Dosage (Solution or Pellet) and MetS Induction Duration	Observed Physiological and Biochemical Parameters for Establishing MetS	Outcomes and Conflicts of Establishing MetS in Animal Model	Remarks and Implications of Study Outcomes
1.	Zemančiková et al. (2016) [33]	Wistar rats and SHR Male 12 weeks old Weight = n/a *n* = 6–10	10% fructose solution for 8 weeks	Physiological: BW; SBP; relative weight of left heart ventricle (LVH), liver, kidney; abdominal fat (rWAT, mWAT, eWAT) Blood biochemistry: GLU, TGL	BW and BP increased significantly in SHR, but not in Wistar rats. Both groups had increased liver weight, GLU, and TGL compared to respective controls	Comparative analysis of SHR to Wistar rats displayed better MetS prognosis in the former, denoting that species selection plays a vital role in being controlled or selected for modelling a disease
2.	Rattanavichit et al. (2016) [34]	SD rats Male, female, OVX female 8 weeks old Male: 260–290 g Female: 180–210 g *n* = 20	10% fructose solution for 6 weeks	Physiological: BW, food and water intake, BP (SBP, DBP, MAP), wet weight (visceral fat and heart) Blood biochemistry: OGTT (plasma GLU and INS), TGL	BW, tissue weight, food and water intake, and INS were significantly different between the sexes and respective controls. INS resistance was not induced in the skeletal muscles of the female and OVX rats, while it was impaired in the male rats	The presence of female reproductive organs (more specifically female hormones) prevented MetS development. Thus, sex needs to be a controlled variable to that ensure pathophysiological outcomes are not misrepresented. This is nullified if the study involves female-specific explorations
3.	Hsieh et al. (2016) [35]	SD rats Male Age = n/a 200–248 g *n* = 8	60% fructose pellets for 16 weeks	Physiological: BW, BP Blood biochemistry: OGTT (GLU and INS), TGL, CHOL, TBARS	BW was visibly increased but was not statistically significantly different. However, BP, GLU, INS, TGL, CHOL, and TBARS increased significantly (*p* < 0.05) in comparison to rats on control chow diet	MetS establishment was successful despite missing BW results, cited as a component in MetS development
4.	Lirio et al. (2016) [36]	Wistar rats and SHR Male 6 weeks old Weight = n/a *n* = 32	10% fructose solution for 6 weeks	Physiological: BW, BP, food and water intake, urine volume, organ weight (heart, lung, liver, kidney), abdominal fat mass (rWAT, mWAT, scWAT) Blood Biochemistry: IPGTT (GLU and INS), total CHOL, HDL, TGL, AST, ALT	Both groups had increased fasting GLU and TGL levels. SHR had lower TGL levels; abdominal mass increased in Wistar rats, but not in SHR. ALT and AST were higher in SHR than in Wistar rats. Unexpectedly, SHR fasting GLU was slightly lower than in Wistar rats	Hyperglycemia, hypertriglyceridemia, and increased abdominal fat mass was comparatively more significant in the Wistar rat model versus the SHR model
5.	Bargut et al. (2017) [20]	CB57BL/6 mice Male 3 months old Weight = n/a *n* = 48	46.43% fructose pellets for 3 weeks	Physiological: BW, BP, food and water intake, eWAT Blood biochemistry: OGTT, TGL, CHOL, INS, HOMA-IR, Adipo, LEP	BP, TGL, CHOL, INS, HOMA-IR, and LEP were increased significantly; Adipo was decreased in the Hfru group compared to control group	Although physical aspects such as BW and fat mass were not statistically significantly differentiated, Hfru induced hypertension, hypertriglyceridemia, hypercholesterolemia, and INS resistance
6.	Bratoeva et al. (2018) [37]	Wistar rats Male Age = n/a 140–180 g *n* = 14	35% fructose solution for 16 weeks	Physiological: BW, organ weight Blood biochemistry: GLU, TGL, TP, Urea, UA, CREA, electrolyte (Na and K), GSH, MDA, CRP	BW, kidney weight, GLUC, TGL, MDA, GSH, UA, and CRP increased significantly	BW increased, hyperglycemia, hypertriglyceridemia, hyperuricemia, and oxidative stress were achieved. The addition of HOMA-IR could be considered, as INS resistance is a notable hallmark of MetS
7.	Ramos et al. (2017) [38]	Wistar rats Male 30 days old 100–148 g *n* = 20	20% fructose solution for 30, 60, and 90 days	Physiological: BW, food and water intake, energy intake, feed efficiency, fat mass (eWAT, mWAT, rWAT), total adipose mass Blood biochemistry: GLU, TGL	Increased BW, fat mass, TGL levels. Total fat mass and eWAT and rWAT deposits were significantly greater in the fructose-fed groups Food and water intake were lower in the Hfru group upon the induction period	Overall, Hfru induced increased BW and adipogenesis, and hypertriglyceridemia Only hyperglycemia was not achieved
8.	Abdelrahman et al. (2018) [21]	Wistar rats Male Age = n/a 180–290 g *n* = 24	60% fructose pellets for 5 weeks	Physiological: BW, BW change, water intake, urine output, BP, heart rate, weight of abdominal aorta Blood biochemistry: GLU, INS, HOMA-IR, TGL, CHOL, HDL, LDL, UA	Water intake and output decreased significantly. Fructose diet incited hypertension, hyperinsulinemia, hypertriglyceridemia, and hypercholesterolemia	MetS was successfully induced in the model, with the exception of BW, which was maintained throughout the experimental phase
9.	Ng et al. (2018) [39]	SD rats Male Age = n/a 180–200 g *n* = 24	60% fructose pellets for 3 and 5 months (FR-3 and FR-5, respectively)	Physiological: BW, tissue weight, BP, urine output Blood biochemistry: Plasma GLU, UA, CHOL, TGL, CREA, electrolytes (Na and K)	BW and UA were significantly higher in both FR-3 and FR-5 as compared to the control only. BP, UA, TGL, and CHOL were higher in FR-5 compared to FR-3 and the control. Conversely, fasting GLU was higher at 3 months than 5 months	Overall, fructose feeding enabled increased BP and proteinuria, but only hypercholesterolemia was significant by 5 months. This study highlights the importance of the duration in inducing MetS to be acceptably significant for research purposes. In comparison to other studies, this duration is deemed overtly lengthy, possibly allowing age to be a confounding factor in MetS development
10.	Chen et al. (2018) [40]	SD rats Male 10 weeks old 200–248 g *n* = 24	60% fructose pellet for 9 weeks	Physiological: BW, BP Blood biochemistry: Plasma GLU, INS, CHOL, TGL	Fructose-fed rats exhibited increased BW, GLUC, and BP (*p* < 0.01). Similarly, INS, CHOL, and TGL levels increased significantly (*p* < 0.001)	Long-term 9-week feeding induced significant levels of obesity, hyperglycemia, hyperinsulinemia, hypertension, and hyperlipidemia
11.	Gambaro et al. (2018) [22]	Swiss mice Male 4 months old Weight = n/a *n* = 30	20% fructose solution for 6 and 10 weeks	Physiological: BW, food and water intake, BP, adipose tissue weight (iWAT, eWAT, rWAT), liver weight Blood biochemistry: GLU, OGTT, AUC, TGL, LEP, ALT, AST	During both durations, BW, AUC, TGL, and LEP were significantly increased; food intake decreased. Hfru plasma LEP and only ALT were significantly higher at 10 weeks than 6 weeks	Food intake was superseded despite being decreased, as Hfru would have larger feed efficiency due to a hypercaloric diet. The results of induction were substantially reliable by week 10. Hyperglycemia was not achieved in either induction period when compared to the control
12.	Subramani et al. (2019) [41]	SD rats Male 3 months old 148–200 g *n* = 34	66% Hfru pellets for 6 weeks	Physiological: BW, body length, BMI, BP Blood biochemistry: Fasting GLU, serum lipid profile (CHOL, TGL, HDL, LDL)	After 6 weeks, Hfru feeding increased BMI, GLU, MAP, TGL, CHOL, and LDL compared to the control. Expectedly, HDL decreased compared to the control	Hfru feeding successfully induced weight gain, hypertension, hyperglycemia, and hyperlipidemia
13.	Fakhoury-Sayegh et al. (2019) [23]	Wistar rats Male 6 weeks old 148–200 g *n* = 40	10%, 20%, and 30% fructose pellets for 16 weeks	Physiological: BW, food intake, energy intake, organ weight (liver, kidney, pancreas, eWAT) Blood biochemistry: Fasting GLU, INS, TGL, ALT, AST	After 16 weeks, serum GLU increased in Hfru groups compared to the control. AST decreased in 20% and 30% Hfru groups only but INS levels were increased in the 10% Hfru group. ALT only increased in the control and 10% Hfru groups	This study utilized the lowest concentrations of fructose in pellet diet paired with the longest induction duration to establish the minimum threshold for NAFLD, a MetS-related disease. For a long-term study, this may perhaps be applicable, but in most studies, it is desirable to achieve effects in an optimal duration
14.	Zhang et al. (2020) [42]	C57BL/6J, DBA/2 J (DBA), FVB/NJ (FVB) mice Male 8 weeks old 20–25 g *n* = 8–12 per strain Control = 8–10 per strain	8% fructose solution for 12 weeks	Physiological: BW; food intake; caloric intake; organ weight (hypothalamus and liver); lean and fat mass of rWAT, mWAT, and scWAT. Blood biochemistry: IPGTT, plasma INS, GLU, lipids	Only DBA mice gained significant BW, fat mass, and fat mass percentage; percentage of lean mass was decreased (increased fat mass ratio); rWAT, mWAT and scWAT weight was increase; GLU homeostasis levels were impaired compared to the C57BL/6J and FVB mice. Plasma INS was significantly elevated in DBA and FVB mice. All groups had increased fructose/water intake and decreased food intake but maintained caloric intake	This study elucidated the comparison between *M. musculus* strains; the final outcome was that DBA mice demonstrated greater susceptibility to chronic fructose ingestion than a SHR model.
15.	Mustafa et al. (2020) [43]	SD rats Male 3 months old 180–200 g *n* = 30	20% (F20%) and 40% fructose (F40%) solution for 6 weeks	Physiological: BW, body length, abdominal circumference, BMI Blood biochemistry: GLU, INS, HOMA-IR, TP, Alb, Gb, urea, CREA, ALT, AST, BIL, lipid profile (TGL, VLDL, HDL, LDL)	BW, BW gained (%), and BMI were statistically significantly increased compared to the control. Urea and CREA were increased significantly in the F40% group only. AST, BIL, TGL, and HDL were reduced; VLDL and LDL were increased significantly compared to the control	The MetS diet-induced model was successful through 40% fructose solution for 6 weeks while 20% fructose solution was insufficient. The animal model demonstrated insulin resistance, obesity, dyslipidemia, metabolic imbalance, and inflammation
16.	Ferreira-Santos et al. (2020) [44]	Wistar rats Male 8 weeks old 225 g *n* = 28	20% fructose solution for 12 weeks	Physiological: BW, caloric intake, organ weight (heart, liver, abdominal fat, mesenteric, aorta artery), tibia length, SBP Blood biochemistry: OGTT, fasting blood GLU, INS, AUC, HOMA-IR, lipid profile (TGL CHOL, HDL, LDL), AST, ALT	SBP, GLUC, AUC, INS, HOMA-IR, abdominal fat mass, TGL, CHOL, ALT, AST, and BMI were increased significantly in the fructose group compared to the control	The Wistar model struggled to gain physical weight although increased organ weight, fat mass accumulation, hyperglycemia, dyslipidemia, and inflammation were achieved
17.	Kitagawa et al. (2020) [45]	Wistar rats Male 4 weeks old Weight = n/a *n* = 13	60% fructose pellets for 2, 4, and 6 weeks	Physiological: BW, organ weight (liver, kidney, heart, biceps femoris, eWAT) Blood biochemistry: HOMA-IR (GLU and INS), TGL, CHOL, FFA	HOMA-IR, TGL, CHOL, and FFA increased during week 2 and continued the increasing trajectory during week 4 and further by 6 weeks of feeding	Weight gain and hyperglycemia were not achieved. However, IR, hyperinsulinemia, and dyslipidemia were achieved in this animal model
18.	Kim et al. (2020) [46]	SD rats Male 11 weeks old 348 g *n* = 12	20% fructose solution for 2 weeks	Physiological: BW, food intake, water intake, BP, urine volume Blood biochemistry: GTT (fasting GLU, AUC), TGL, CHOL	BW, daily water intake, BP, GLUC, AUC, and water retention increased significantly after 2 weeks of 20% fructose. Urine volume was decreased compared to the control	The SD rat model achieved MetS. However, the food and water intake did not differ. The 2-week induction duration is considerably short and may be a factor

Abbreviations: SHR, spontaneously hypertensive rats; SD, Sprague-Dawley; OVX, ovariectomized; Hfru, high-fructose; BW, body weight; HOMA-IR, homeostasis model assessment of insulin resistance; BMI, body mass index; SBP, systolic blood pressure; DBP; diastolic blood pressure; MAP, mean arterial pressure; OGTT, oral glucose tolerance test; IPGTT, intraperitoneal glucose tolerance test; GLU, glucose; INS, insulin; AUC, area under the curve; TGL, triglyceride; CHOL, cholesterol; HDL, high-density lipoproteins; LDL, low-density lipoproteins; VLDL, very low-density lipoproteins; FFA, free fatty acids; rWAT, retroperitoneal white adipose tissue; mWAT, mesenteric white adipose tissue; scWAT, subcutaneous white adipose tissue; Alb, albumin; Gb, globulin; AST, aspartate aminotransferase; ALT, alanine aminotransferase; Adipo, adiponectin; LEP, leptin; TP, total protein; BIL, total bilirubin; UA, uric acid; CREA, creatinine; Na, sodium; K, potassium; GSH, glutathione; MDA, malondialdehyde; CRP, C-reactive protein; TBARS, thiobarbituric acid-reactive substances.

**Table 2 nutrients-13-02497-t002:** Additional biological parameters, methodology, and outcomes of MetS and related diseases.

No.	First author, Year of Publication, and Main Study Objective (Cardiac, Diabetes, Bone, Cancer)	Additional Biological Parameters (Inflammatory Markers, Genes, Proteins)	Outcomes of Additional Biological Parameters
1.	Zemančiková et al. (2016) [33] MetS and cardiovascular function	Isometric tension endothelium-dependent vasorelaxation, adrenergic contractions in endothelium-intact mesenteric arteries	Impaired relaxation of the aorta in both SHR and Wistar rats but not in nitric oxide (NO)-deficient Wistar rats. Diminution of adrenergic contractions was observed in SHR mesenteric arteries, but not in the Wistar rat counterparts. Contractile response in K^+^ concentrated bath solution was not altered between groups
2.	Rattanavichit et al. (2016) [34] MetS and ovarian hormone function	Wet weight of uterus and skeletal muscle (soleus). INS-mediated muscle GLU transport activity. Analysis of signaling elements in skeletal muscle (IRS-1 Tyr989, IRS-1 Ser307, Akt Ser451, AS160 Ser588, JNK Thr183/Tyr185, p38 MAPK Thr180/Tyr182)	INS resistance was not induced in the skeletal muscles of non-OVX and OVX female rats, while male rats exhibited the following changes: Decreased --> IRS-1 Tyr989 (34%), Akt Ser451 (30%), AS160 Ser588 (43%) Increased --> IRS-1 Ser307 (78%), JNK Thr183/Tyr185 (69%), p38 MAPK Thr180/Tyr182 (81%) The above genetic analysis shows that male rats had significantly impaired INS-mediated GLU uptake, leading to loss of skeletal muscle mass and mitochondrial activity in energy production
3.	Hsieh et al. (2016) [35] MetS and liver function	2D gel-based proteomics. MALDI-TOF/MS-MS to identify hepatic protein expression patterns in rat liver	Analysis identified proteins that were: Increased --> FAS, 78-kDa G-RP, LAT1, triokinase, GFAP, FBP-1, MAAI, GSTA3, PRDX1 Decreased --> ACSM1, GDI-1 These results indicate the impaired FA metabolism onset of carcinogenicity, inflammatory mediators (FAS), and metabolic dysfunction originating from the liver
4.	Lirio et al. (2016) [36] MetS and NAFLD	Histology of liver tissue	There was interstitial fat deposition and fibrosis in the liver of SHR only
5.	Bargut et al. (2017) [20] MetS and WAT	Histology of eWAT (adipocyte area), RT-qPCR, Western blotting	Hfru-treated rats displayed adipocyte area increase by 21%, implying hypertrophy, inflammation, and uncontrolled lipolysis compared to control groups Inflammatory markers: Increased --> TNF-α, IL-6, F4/80, MCP1, pERK, pJNK, NF-κB Anti-inflammatory markers: Increased --> ATGL, HSL, pHSL, β3-AR, CD36, aP2 Decreased --> Adiponectin, PPAR-γ gene, perilipin, pAMPK, pAKT No change --> IL-10 gene and PPAR-γ protein
6.	Bratoeva et al. (2018) [37] MetS and renal function	Histological study of the kidney	Proximal and distal section of the tubular region revealed signs of vacuolization and degeneration, respectively. Vascular rupture, scarring, and atherosclerotic manifestations were apparent, but not in the control. The glomeruli and subendothelial layers of blood vessels had nodule formation, shown by positive amyloid stains. These traits show the loss of glomeruli function and vascular deterioration that matched MetS symptoms
7.	Ramos et al. (2017) [38] MetS	No additional parameters were observed	No additional findings were made
8.	Abdelrahman et al. (2018) [21] MetS and inflammatory and oxidative markers	Inflammatory markers, oxidative stress and NO in fructose-fed rats: Aortic endothelin-1, plasma and aortic NO, plasma TNF-α, IL-6, plasma superoxide dismutase (SOD), plasma catalase, plasma MDA. Abdominal aorta removed and weighed	When compared to the control, the Hfru group displayed significant levels of: Increased --> MDA, TNF-α, IL-6, aortic NO, endothelin-1 Reduced --> Catalase, SOD These results complement the biochemical parameters that had established progressive inflammation, oxidative damage, and hypertensive disorder
9.	Ng et al. (2018) [39] MetS and renal function	Gene expression study and immunohistochemistry of the liver and kidney	The FR-5 group expressed lower GLUT1, compensated by increased GLUT2 and GLUT9 compared to the FR-3 group. Both groups had equally increased SGLT1, SGLT2, renal UA expression, and FA deposition. Increased GLUT9 and UA are correlated to signs of hyperuricemia; SGLT1:SGLT2 imbalance is identified as ineffective sodium–glucose reabsorption function of the kidney. These findings also describe the loss of INS function compensated by SGLT2 INS-independent function of GLU reabsorption
10.	Chen et al. (2018) [40] MetS and bladder function	Rat bladder was processed into individual muscle strips for organ bath pharmacological studies. RT-PCR and Western blot analysis of bladder control via genetic and protein expressions of cannabinoid receptor (CB)1/CB2 receptors	Acetylcholine bladder muscle strip contraction showed statistically insignificant differences between Hfru and control groups. Western blotting showed decreased CB1/CB2 protein levels following MetS diet induction. Similarly, RT-PCR showed that CB1/CB2 mRNA was decreased significantly in the fructose group. These outcomes signify the loss of the homeostatic role of CB1 and CB2, affecting the parasympathetic stimulus for bladder function, as observed in the MetS model
11.	Gambaro et al. (2018) [22] MetS and adipocyte function	Metabolic mRNA analysis of adipose tissue. Histological analysis of adipose tissue	In both periods (6 and 10 weeks), there was increased eWAT; decreased adiponectin, IL-6, IL-10, TNF-α, CD206; and hypertrophy and parenchymatous degeneration of hepatocytes. Only the 10-week group had decreased Il-10 and Il-1b, with instances of lipid accumulation foci in histological sections compared to the 6-week group. These findings show that the accumulation of fat mass increased proportionally to the feeding duration. It stimulated inflammatory macrophages (M1) and decreased the anti-inflammatory response, leading to liver damage and loss of function in the diet-induced MetS model
12.	Subramani et al. (2019) [41] MetS	Oxidative stress measurement (TBARS, SOD). Rat nuclear factor kappa B (NF-κB). Histopathology of the liver and heart	The Hfru group had significantly increased TBARS and decreased SOD but non-significantly increased NF-κB compared to the control diet groups. The Hfru group had fatty infiltration, with micro- and macrovesicular hepatic steatosis and hypertrophied heart compared to the control groups. These results show the effect of MetS that leads to a hypertrophic heart to circulate and compensate for atherosclerosis and other loss of function. Fructose feeding accumulated fatty deposition and ROS and inflammatory cytokine secretion, leading to dyslipidemia of the liver
13.	Fakhoury-Sayegh et al. (2019) [23] MetS with hepatic, renal, and pancreatic function	Oxidative stress (MDA). Serum adiponectin levels. Inflammatory markers (TNF-α and IL-6). Histopathology examinations of the liver, kidney, pancreas, and epididymal fat pads	Adiponectin levels decreased in group 2 and 3 (20% and 30% Hfru, respectively). MDA levels decreased in all groups except group 2. TGL, TNF-α, and IL-6 were not significant between the fructose groups and control after 16 weeks. The control and group 2 and 3 had minor signs of microvesicular steatosis. Only group 4 (30% Hfru) presented milder microvacuolar steatosis. Necroinflammation and fibrosis were absent from all groups, but perisinusoidal fibrosis was apparent (>20%) in group 3 and 4. Group 3 and 4 kidneys showed mild renal inflammation (>60%) and lower glomerulosclerosis and interstitial renal fibrosis. The pancreas did not show hyperplasia or hypertrophy of the islets of Langerhans. Islet distribution, size, and shape were not altered or fibrotic. After 16-week Hfru intervention, the onset of liver fibrosis and kidney failure were proportional to Hfru dosage. INS function and histopathology of the pancreas were unchanged, thus remaining functional
14.	Zhang et al. (2020) [42] MetS and genetically different *M. musculus* strains	RNA sequencing of the hypothalamus, liver, and mWAT to determine the key drivers in liver function to validate strain individuality properties. Subsequently, deduce differentially expressed genes (DEGs) in the hypothalamus, liver, and WATs	Strain-specific DEGs were discovered, as DBA mice represented the largest volumes in all three tissues for lipid metabolism. However, some overlapping DEGs did not show strain specificity but were affected by fructose metabolism. The categories were: Xenobiotic stimulus (*Gstp1*, *Ephx1*, *Gstm1*) Organic cyclic compounds (*Abca1*, *Id3*, *Abat*) Metabolic processes (*Abca1*, *Htatip2*, *Grhpr*, *Nudt7*) Transcriptional regulation (*Ier5*, *Jun*, *Id3*) Immune modulation (*Cd200*)
15.	Mustafa et al. (2020) [43] MetS	No additional parameters were observed	No additional findings were made
16.	Ferreira-Santos et al. (2020) [44] MetS	Morphometric determinations (LVH, liver, adiposity measurement). Histological studies (liver and rWAT). Vascular reactivity. Oxidative stress (superoxide anion and lipid peroxidation)	LVH, liver index, oxidative stress, and adiposity increased significantly in the fructose group compared to the control. Histological staining showed morphological alterations in hepatocytes and adipocytes. The liver of fructose-fed rats had lipid accumulation. Endothelial dysfunction manifested from the reduced relaxation in response to acetylcholine (Ach) in the aortic and mesenteric rings of the fructose group. A Hfru diet facilitated liver steatosis and hypertension due to stiff endothelial contraction and lipid-derived oxidative damage to metabolic organs
17.	Kitagawa et al. (2020) [45] MetS	Serum vitamin E; relative vitamin E and lipid peroxide (LPO) concentrations of the heart, liver, kidney, skeletal muscle (SM); WAT	Liver, kidney, heart, and SM vitamin E content were significantly higher in the fructose group than in the control by 2 weeks. However, vitamin E in WAT was significantly lower in the fructose group, which continued on a downward trend until 6 weeks. Heart, WAT, and liver LPO increased and achieved statistical significance by 6 weeks. Vitamin E plays a pivotal role in antioxidative responses, which was hampered by MetS induced by a Hfru diet. This accrued damage in the heart, liver, and organs surrounded by WAT
18.	Kim et al. (2020) [46] MetS and adrenal function	Hormone levels (serum renin, angiotensin [Ang] II, aldosterone). Histological analysis of the kidney and adrenal gland. RT-qPCR (*Ren*, *At1ar*, *At1br*, *Ace*, *At2*, *Agt*, *Gapdh*)	Serum levels of renin and Ang II were significantly elevated by fructose supplementation. Fructose intake increased expression of *Agt* in the liver and *Ace* in the lungs. Fructose intake increased AT1R and Ang I protein levels in the kidney. Histological examination showed no significant effect on collagen deposition and fibrosis, which would have appeared blue upon trichrome staining. The fluctuation of the adrenal metabolites above affected the lung, kidney, and liver functions. Hence, the fructose diet disrupted pulmonary, hepatic, and renal function by altering the adrenal maintenance of vascular properties crucial for biochemical homeostasis and nutrient supplementation

**Table 3 nutrients-13-02497-t003:** Durations of MetS induction to physiological and biochemical outcomes of MetS.

Duration of MetS Induction (weeks)	References of Study(s)	Physiological and Biochemical Parameters of MetS						
		Increased BW, Length or Abdominal Circumference	Increased Organ or Tissue Mass	Increased Caloric Intake (Hyperphagia, Hyperleptinemia)	High Blood Pressure (Hypertension)	High Blood Glucose Levels (Hyperglycemia)	Dyslipidemia (Hypertriglyceridemia, Hypercholesterolemia)	Hyperinsulinemia or Insulin Resistance
2	[46,45]	/	/	X	O	O	O	/
3	[20]	X	X	O	/	O	/	/
4	[38,45]	/	O	O	O	O	/	/
5	[21]	X	O	O	/	O	/	/
6	[22,34,36,41,43,45]	/	/	/	/	/	/	/
8	[33,38]	/	/	O	/	/	/	/
10	[22]	/	O	O	O	X	/	/
12	[38,39,42,44]	/	/	X	O	X	O	/
13	[39]	/	O	O	/	O	/	O
16	[23,35,37]	X	X	X	/	/	/	/
21	[39]	/	/	O	O	X	/	O

(/) = statistically significant results, (*p* < 0.05); (X) = statistically non-significant results (*p* > 0.05); (O) = test parameter unavailable or not assessed in study(s).

**Table 4 nutrients-13-02497-t004:** Concentration of types of fructose diet to physiological and biochemical outcomes of MetS.

Conc. of Fructose (%)	References of Study(s)		Physiological and Biochemical Parameters of MetS						
	Solution	Pellet	Increased BW, Length or Abdominal Circumference	Increased Organ or Tissue Mass	Increased Caloric Intake (Hyperphagia, Hyperleptinemia)	High Blood Pressure (Hypertension)	High Blood Glucose Levels (Hyperglycemia)	Dyslipidemia (Hypertriglyceridemia, Hypercholesterolemia)	Hyperinsulinemia or Insulin Resistance
8	[42]	-	/	/	X	O	X	O	O
10	[33,34,36]	[23]	/	O	/	/	O	/	/
20	[22,38]	[23]	/	/	X	O	O	/	O
30	-	[23]	X	X	X	O	/	/	/
40	[43]	-	/	O	O	O	/	/	/
45.43	-	[20]	X	X	O	/	O	/	/
60	-	[21,35,39,40,41,45]	X	/	/	/	/	/	/
66	-	[41]	/	O	O	/	/	/	O

(/) = statistically significant results, (*p* < 0.05); (X) = statistically non-significant results (*p* > 0.05); (O) = test parameter unavailable or not assessed in study(s).

## Data Availability

Not applicable.

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
