# Peer review of "Recent Developments in Rodent Models of High-Fructose Diet-Induced Metabolic Syndrome: A Systematic Review"

_nutrients, 2021, doi:10.3390/nu13082497_

Round 1

Reviewer 1 Report

The authors have responded to my requests.

Author Response

The authors have responded to my requests.

Answer: Thank you for acknowledging our answers to corrections. 

Reviewer 2 Report

The authors have answered to my comments. However there are still a few details to be corrected. page 14, line 283-284: OVX is not the abbreviation for non-ovariectomized animals. Please correct. page 14, line 289: please remove the abbreviation GLU once it is used for glucose and related hexoses transporters thus it makes the sentence confusing.

Author Response

Point 1: page 14, line 283-284: OVX is not the abbreviation for non-ovariectomized animals. Please correct.

Answer: Thank you for noticing the mistake. We have removed the OVX abbreviation for non-ovariectomized animal [Line 271].

Point 2:  page 14, line 289: please remove the abbreviation GLU once it is used for glucose and related hexoses transporters thus it makes the sentence confusing.

Answer: Thank you for your attention. We have removed the abbreviation as recommended to preserve meaning. [Line 276]

This manuscript is a resubmission of an earlier submission. The following is a list of the peer review reports and author responses from that submission.

Round 1

Reviewer 1 Report

This systematic review will be of interest to specialists in the area of metabolic syndrome. The information presented is abundant and the bibliography is well selected. Only minor comments:

1) The authors claim "First screening of title and abstract were performed according to the inclusion and exclusion criteria". The inclusion and exclusion criteria should be detailed and clarified to the reader.

2) The text in the tables should be shortened and try to be outlined with shorter sentences and perhaps use signs and arrows.

3) Limitations regarding the sex of animal models should also be included in the conclusions. 

Reviewer 2 Report

The authors presented a review focused on studies of diet-induced obesity in rodent models after fructose consumption. However major and minor points need to be corrected. Please revise carefully the metabolic steps and respective enzymes and make clear some aspects related with those steps. Please also re-write part of the « Remarks and implications on outcomes of the study » mainly from study 1 to 8.

Comments and suggestions for authors :

Please make consistent the designation Type or type (when referring to Type I or Type II Diabetes.

When two or more references are added to the text, please leave a space between the numbers

Figures or Fig. should be uniformed

Please make the abbreviations consistent along the overall manuscript and figure captions or tables.

page 2, line 81 : « metabolic diseases », not « auto-immune diseases ». please correct. The mentioned diseases are metabolic diseases.

page 3, line 106 : Does the sentence mean « one of the three most common sugars » ?

page 3, lines 106-107 : Please clarify the meaning of « metabolized directly ». Fructose is rapidly phosphorylated to fructose-1-phosphate by fructokinase which is then cleaved into DHAP and glyceraldeyde which is further phosphorylated to glyceraldeyde-3-phosphate. Do you mean that fructose is not metabolized directly to glycogen like glucose ? Glycogen can be synthesized by the direct pathway (from an intact glucose molecule) or by the indirect pathway (from non-carbohydrate precursors). Either DHAP and glyceraldeyde-3-phosphate are intermediates of the indirect pathway (gluconeogenic pathway). Please also clarify the meaning of « nor as immediately » once fructose is rapidly metabolized to fructose-1-phosphate without any control.

page 3, line 109 : « both products are stored as fats ». Could you please clarify what products are you referring to ? Please rephrase the sentence because it seems that you say that glycogen and triglycerides are stored as fat. Please also clarify the next sentence (line 110).

page 3, line 124 : Please highlight that it is the intestinal lumen.

Figure 1 : Although there are common intermediate metabolites for glycolysis and fructolysis, please highlight the fructose metabolic pathways once the title is fructolysis

page 4, line 135 : CO2 ( 2 should be as subscript)

page 4, line 136 : I suggest to avoid the designation « by-products » for the mentioned molecules.

page 4, line 137-138 : Phosphofructokinase does not cleave fructose-6-phosphate into fructose-1,6-bisphosphate. That enzyme is a key regulatory step in the glycolytic pathway and phosphorylates fructose-6-phosphate to fructose-1,6-bisphosphate. This must be corrected. Please also briefly describe the role of citrate and ATP on activity of this enzyme.

page 4, lines 138-143 : please clarify the sentences. They are quite confusing.

page 4, lines 144 : What does « upstream regulations are neutralized... » mean ? There are no upstream regulations.

page 4, line 158 : TNFα, not TNFa. Please correct.

Figure 2 : A parentheses is missing for Methods and Results.

page 6, line 223-224 : The number of the articles searched do not match with those of figure 2. Please correct.

page 6, line 227-228 : Please clarify the meaning of the sentence « ...Although...not identical »

Table 1 : please add the references of the studies.

Table 1 : n=n/a means that the number of animals included in the referred study is not known ? This is an important data.

Table 1 : Please keep the designations constant. As an example, for study 4. (Lirio et al. 2016) either is mentioned Wistar and SHR rat or normotensive or hypertensive rats. Using different designations make the text difficult to read.

In study 2. (Rattanavichit et al. 2016), the meaning in OXC-females is not explained.

Table 1 : Triglycerides appear as TG, TGL, tg or in full. Please uniformed. The same for the other metabolites and conditions such as blood pressure. Please also check and uniform fructose-fed, HFru, FRD. Please check carefully for all the studies its description, abbreviations and upper and lower case.

Table 1 : Statistically null should read statistically not significantly different.

Table 1 : Study 8, (Abdelrahman et al. 2018) the sentence « Body weight was not significantly different » should read statistically not significantly different.

Table 1 : General comment : Table 1 must be improved.

page 14, line 264 : Does B6 mean C57BL/6 ? please uniform the designation.

page 14-15, line 271-273 : The sentence about the OVX group in unclear. Please rephrase and make clear that ovariectomized female rats are in disadvantage relatively to non- ovariectomized female rats. Also rephrase the description of the results obtained from this study (study 2).

Table 2 : Study 3-outcomes of additional biological parameters : onset inflammatory mediators (FAS) should be defined once FAS usually refers to fatty acid synthase. Does FAS mean Fas ?

Table 2 : Like for Table 1, please add the references of the studies

Table 2 : study 9- the designation of FR-5 or FR-3 should match with those of Table 1. All the designations should be consistent for both tables.

Table 1 and 2 : study 6-The authors referred Bratoeva et al. 2017 however in the reference list reads Bratoeva et al. 2018.

Table 3 and Table 4 : Please add the references of the studies

Reviewer 3 Report

I've read with attention the paper of Alvin Man Lung Chan et al. that aimed to review and compare the previous studies of metabolic syndrome performed on rodent models. The study concerned important aspects for planning future preclinical studies. Overall, the manuscript is logically organized and sufficiently described. The methodology applied is correct. The results were correctly presented in the form of a readable table.

In the following, I listed my comment to improve the manuscript (MS):

1.  Line 75, 83, 163, 244, and through the entire manuscript -> No consistency in using the abbreviation. The abbreviation should be inserted on first use. Please review and correct the manuscript for abbreviations. Similarly, please be consistent in using the MetS abbreviations.2.  Line 69 and 72 -> inconsistent record: Type 1 Diabetes or diabetes mellitus type II. 3.  Line 106, -> I suggest changing the reference notation from [11, 12, 13] to  [11-13] . Similar situation in lines 143, 160, 172, 249, 2524.  Line 181 -> Please correct the reference notation5.  Table 2, line 2 -> typo 6.  Overall, the English is acceptable, but to raise the perception of this review I suggest improving the English language with the help of a native speaker.